# MLKL and CaMKII Are Involved in RIPK3-Mediated Smooth Muscle Cell Necroptosis

**DOI:** 10.3390/cells10092397

**Published:** 2021-09-12

**Authors:** Ting Zhou, Elise DeRoo, Huan Yang, Amelia Stranz, Qiwei Wang, Roman Ginnan, Harold A. Singer, Bo Liu

**Affiliations:** 1Department of Surgery, School of Medicine and Public Health, University of Wisconsin-Madison, Madison, WI 53705, USA; zhout@surgery.wisc.edu (T.Z.); ederoo@wisc.edu (E.D.); yangh@surgery.wisc.edu (H.Y.); astranz@wisc.edu (A.S.); Qiwei_Wang@dfci.harvard.edu (Q.W.); 2Department of Molecular and Cellular Physiology, Albany Medical College, Albany, NY 12208, USA; ginnanr@amc.edu (R.G.); singerh@amc.edu (H.A.S.); 3Department of Cellular and Regenerative Biology, School of Medicine and Public Health, University of Wisconsin-Madison, Madison, WI 53705, USA

**Keywords:** necroptosis, smooth muscle cells, MLKL, CaMKII, abdominal aortic aneurysm

## Abstract

Receptor interacting protein kinase 3 (RIPK3)-mediated smooth muscle cell (SMC) necroptosis has been shown to contribute to the pathogenesis of abdominal aortic aneurysms (AAAs). However, the signaling steps downstream from RIPK3 during SMC necroptosis remain unknown. In this study, the roles of mixed lineage kinase domain-like pseudokinase (MLKL) and calcium/calmodulin-dependent protein kinase II (CaMKII) in SMC necroptosis were investigated. We found that both MLKL and CaMKII were phosphorylated in SMCs in a murine CaCl_2_-driven model of AAA and that *Ripk3* deficiency reduced the phosphorylation of MLKL and CaMKII. In vitro, mouse aortic SMCs were treated with tumor necrosis factor α (TNFα) plus Z-VAD-FMK (zVAD) to induce necroptosis. Our data showed that both MLKL and CaMKII were phosphorylated after TNFα plus zVAD treatment in a time-dependent manner. SiRNA silencing of *Mlkl*-diminished cell death and administration of the CaMKII inhibitor myristoylated autocamtide-2-related inhibitory peptide (Myr-AIP) or siRNAs against *Camk2d* partially inhibited necroptosis. Moreover, knocking down *Mlkl* decreased CaMKII phosphorylation, but silencing *Camk2d* did not affect phosphorylation, oligomerization, or trafficking of MLKL. Together, our results indicate that both MLKL and CaMKII are involved in RIPK3-mediated SMC necroptosis, and that MLKL is likely upstream of CaMKII in this process.

## 1. Introduction

Smooth muscle cell (SMC) death plays a significant role in the pathophysiology of abdominal aortic aneurysm (AAA) [1], a relatively common aortic disease characterized by progressive expansion and in some cases life-threatening rupture of the abdominal aorta. Our lab has previously shown that receptor interacting protein kinase 3 (RIPK3)-mediated SMC necroptosis contributes to AAA pathology [2]. Briefly, necroptosis is a programmed form of necrotic cell death distinct from necrosis and apoptosis. In the setting of inflammatory stimuli, RIPK3 and its signaling partner receptor interacting protein kinase 1 (RIPK1) become phosphorylated and lead to necroptosis [3]. Our lab has demonstrated that aortic tissue from AAA patients have elevated levels of RIPK3 [2]. In mice, inhibition of necroptosis either by gene deletion of *Ripk3* or by chemical inhibitors to RIPK1 or both RIPK1 and RIPK3 prevents medial SMC depletion and reduces aortic dilatation [2,4,5].

Mixed lineage kinase domain-like pseudokinase (MLKL) is the most well-established substrate of RIPK3 [6]. Phosphorylation, oligomerization, and membrane translocation of MLKL are necessary steps for the execution of necroptosis in many cell types [7,8]. However, Zhang et al. reported that during ischemia-reperfusion or doxorubicin-induced myocardial necroptosis, RIPK3 may signal through calcium/calmodulin-dependent protein kinase II (CaMKII) instead of MLKL [9]. Activation of CaMKII was also detected in vinblastine or tunicamycin-induced myocardial necroptosis, and in Bisphenol A-induced coronary endothelial cell necroptosis [10,11,12]. The precise roles of MLKL and CaMKII in SMC necroptosis remain elusive.

The aim of this study was to examine the role of MLKL and CaMKII in SMC necroptosis. We found that both MLKL and CaMKII were phosphorylated in aortic tissue from mice with CaCl_2_-induced AAAs and in aortic SMC undergoing necroptosis after tumor necrosis factor α (TNFα) plus Z-VAD-FMK (zVAD) treatment. *Ripk3* deficiency in vivo decreased MLKL and CaMKII phosphorylation. In the in vitro setting, inhibition of MLKL or CaMKII expression reduced SMC death. Further delineation of the signaling pathway suggested that both MLKL and CaMKII participate in RIPK3-mediated SMC necroptosis, with MLKL being upstream of CaMKII.

## 2. Materials and Methods

The data that support the findings of this study are available from the corresponding author upon reasonable request.

### 2.1. Reagents

DMEM and cell culture reagents were purchased from Thermo Scientific (Waltham, MA, USA). Recombinant mouse TNFα was purchased from R&D Systems (Minneapolis, MN, USA. Catalog No. 410-MT). zVAD was purchased from BACHEM (Torrance, CA, USA. Catalog No. N-1510). Myristoylated autocamtide-2-related inhibitory peptide (Myr-AIP) was purchased from Enzo (Farmingdale, NY, USA. Catalog No. ALX-151-030).

### 2.2. Animal Studies

All animal studies conformed to the National Institutes of Health Guide for the Care and Use of Laboratory Animals and were performed with the approval of the Institute Animal Care and Use Committee at University of Wisconsin-Madison (Protocol No. M005792). All experiments were performed on male mice, as AAAs predominantly affect men [13]. All mice had free access to standard laboratory diet (2018 Teklad global 18% protein rodent diets, ENVIGO, Indianapolis, IN, USA) and water. *Ripk3*^+/−^ mice [14] on a C57BL/6 background were generously provided by Dr. Vishva M. Dixit (Genentech) and maintained on heterozygous breeding; *Ripk3*^+/+^ and *Ripk3*^−/−^ littermates were used for experiments.

### 2.3. Modified CaCl_2_ AAA Model

Twelve-week-old male mice (3 mice per group for phospho-MLKL staining, and 4 mice per group for phospho-CaMKII staining) were anesthetized by isoflurane inhalation. The abdominal aorta between the renal and iliac arteries was isolated following a midline incision. A small piece of gauze soaked in 0.5 M CaCl_2_ was applied perivascularly for 10 min. The gauze was replaced with another piece of phosphate buffered saline (PBS)-soaked gauze for 5 min. Mice in sham group received 0.5 M NaCl treatment for 10 min followed by PBS-soaked gauze for 5 min. The incision was closed and 2% lidocaine topical ointment was applied to the suture site. A total amount of 0.5 mg/kg sustained-release buprenorphine was administered subcutaneously before surgery [15]. Four days after AAA induction, mice were euthanized and perfused with PBS, and injured aortas were embedded with optimal cutting temperature compound (Sakura Tissue Tek) and cut into 6 μm cross sections for immunofluorescent staining. 

### 2.4. Immunofluorescent Staining 

Tissue sections were fixed with 4% paraformaldehyde (PFA) for 10 min, followed by permeabilization with 0.1% Triton X-100 for 10 min at room temperature. Non-specific sites were blocked using 3% bovine serum albumin (BSA) for 1 h at room temperature. Tissue sections were incubated at 4 °C overnight in 0.3% BSA and 0.1% Tween 20 containing primary antibodies anti-phospho-MLKL Ser345 (1:1000, Cell Signaling Technology, Danvers, MA, USA, 37333) or anti-phospho-CaMKII Thr287 (1:100, generated and validated by Dr. Singer’s lab) [16,17] and FITC Anti-alpha smooth muscle actin (1:500, Abcam, Waltham, MA, USA, ab8211). Normal rabbit IgG (Thermo Scientific, Waltham, MA, USA, 026102) was used for the negative controls. After washing three times in PBS, sections were incubated with Alexa Fluor 594-conjugated secondary antibodies (1:500, Thermo Scientific, Waltham, MA, USA) for 1 h at room temperature in the dark. DAPI was used to stain nuclei. Images were acquired with a Nikon A1RS confocal microscope system, and analyzed using ImageJ software. 

### 2.5. Cell Culture

Mouse aortic smooth muscle cell line MOVAS cells were obtained from American Type Culture Collection (ATCC, Manassas, VA, USA, CRL-2797) and grown as recommended in DMEM modified containing 4.5 g/L D-Glucose (Thermo Scientific, 11965118) supplemented with 10% fetal bovine serum (FBS), 100 U/mL penicillin, and 100 U/mL streptomycin.

### 2.6. siRNA-Mediated Knock-Down

MOVAS cells were transfected with scrambled siRNA and siRNA against mouse *Mlkl* (Integrated DNA Technologies, Coralville, IA, USA, mm.Ri.Mlkl.13) or *Camk2d* (Qiagen, Germantown, MD, USA, 1027416) using Opti-MEM (Thermo Scientific, 51985034) and Lipofectamine RNAiMAX (Thermo Scientific, 13778150) and incubated for 48 h.

### 2.7. Flow Cytometric Analysis

Cell death was evaluated by using an Annexin V-PE/7-AAD staining Kit (BD Biosciences, San Jose, CA, USA, 559763). Cells were detached with accutase (Innovative Cell Technologies, San Diego, CA, USA, AT104) and centrifuged at 2000 rpm for 5 min. Cell pellets were washed with PBS and resuspended in 100 μL binding buffer. Totals of 5 μL of Annexin V-PE and 5 μL of 7-AAD were added to the cells and incubated at room temperature for 15 min in the dark. After incubation, 400 μL binding buffer was added to each sample. A total of 10,000 cells per sample were analyzed using a Becton Dickinson Biosciences FACSCalibur. 

### 2.8. Western Blotting

Cells were lysed in RIPA buffer (Sigma-Aldrich, St. Louis, MO, USA, R0278) containing protease and phosphatase inhibitors (Halt Cocktail, Thermo Scientific, 78444). To detect MLKL or phosphor-MLKL oligomerization, cells were lysed in 1% digitonin (Thermo Scientific, BN2006) with protease and phosphatase inhibitors. Equal amounts of protein extract were loaded and separated by SDS-PAGE and then transferred to polyvinylidene fluoride (PVDF) membranes. The membranes were blocked for 60 min at room temperature with 5% low fat milk in Tris-buffered saline plus 0.05% Tween 20 (TBST), and then incubated with primary antibodies anti-MLKL (1:1000, Cell Signaling Technology, Danvers, MA, USA, 37705), anti-phospho-MLKL Ser345 (1:1000, Cell Signaling Technology, 37333), anti-pan-cadherin (1:1000, Abcam, ab6528), anti-GAPDH (1:1000, Cell Signaling Technology, 2118), anti-RIPK3 (1:1000, Cell Signaling Technology, 95702), anti-RIPK1 (1:1000, BD Biosciences, San Jose, CA, USA, 610459), anti-β-actin (1:8000, Sigma-Aldrich, A5441), anti-pan-CaMKII (generated and validated by Dr. Singer’s lab) [18,19,20] and anti-phospho-CaMKII Thr287 overnight at 4 °C, followed by HRP-labeled secondary antibodies. Labeled proteins were visualized with an enhanced chemiluminescence system (Thermo Scientific, 34096) and ImageQuant LAS 4000 Mini (GE Healthcare Bio-Sciences, Marlborough, MA, USA). For quantification, optical densities of proteins were determined by ImageJ.

### 2.9. Subcellular Fractionation

Cells were fractionated into cytoplasmic and membrane fractions [21]. Cells were permeabilized in buffer (20 mM HEPES pH 7.5, 100 mM KCl, 2.5 mM MgCl_2_ and 100 mM sucrose, 0.025% digitonin, 2 μM N-ethyl maleimide, phosphatase, and protease inhibitors). Crude membrane and cytoplasmic fractions were separated by centrifugation at 11,000× *g* for 5 min. The supernatant was the cytoplasmic fraction. Cell pellet was further solubilized in permeabilization buffer + 1% digitonin and clarified by centrifugation at 20,000× *g* for 30 min. The supernatant from this step was the membrane fraction. Digitonin was added to the cytosolic fraction to 1%, and both cytosolic and membrane fractions were subjected to immunoblotting.

### 2.10. Co-Immunoprecipitation

Cells were lysed in Pierce IP Lysis Buffer (Thermo Scientific, 87787) then co-immunoprecipitation experiments were performed using the SureBeads magnetic beads (Biorad, 1614013) according to the manufacturer’s protocol. In brief, Protein A magnetic beads were washed in TBST then incubated with anti-RIPK3 (Cell Signaling Technology, 95702), anti-MLKL (Cell Signaling Technology, 37705), or anti-pan-CaMKII antibodies or rabbit IgG isotype control (Thermo Scientific, 026102) for 30 min at room temperature. Beads were magnetized and washed 3 times with TBST, then incubated with cell lysate overnight at 4 °C. After incubation, beads were washed 3 times with TBST, and immunoprecipitated proteins were eluted in 1× Laemmli buffer and subjected to immunoblotting.

### 2.11. Statistical Analysis

Results are presented as mean±SD. Data were assessed for normality using the Shapiro–Wilk normality test. Data not exhibiting a normal distribution were log2-transformed and retested for normality. One-way ANOVA with Tukey post hoc test for normally distributed data and Kruskal–Wallis nonparametric test for skewed data after transformation were used to compare ≥3 means. Two-way ANOVA followed by Sidak multiple comparisons were performed to compare how a response is affected by 2 factors. Statistical analyses were performed with GraphPad Prism 7.0 (GraphPad Software, Inc). Experiments were repeated as indicated. Differences with *p* < 0.05 were considered statistically significant.

## 3. Results

### 3.1. MLKL and CaMKII Are Both Activated in Aneurysmal Aortic Tissues

Using antibodies specific to phosphorylated MLKL or CaMKII, we detected activation of both MLKL and CaMKII in CaCl_2_-induced mouse aortic aneurysm tissue. Phospho-MLKL was nearly undetectable in sham controls, but became prominent in aortic tissues subjected to aneurysm induction, almost exclusively in medial SMCs (Figure 1A,B). In contrast, noticeable levels of phospho-CaMKII were found in sham tissues and in cells outside of the medial layer (Figure 1C). Aneurysm induction increased phospho-CaMKII accumulation in medial SMCs by ~2 fold (Figure 1C,D). Ripk3 deficiency markedly reduced the aneurysm-induced upregulation of phospho-MLKL and phospho-CaMKII (Figure 1), demonstrating the dependence of MLKL and CaMKII activation on RIPK3.

To delineate the function of MLKL and CaMKII in SMC necroptosis, we turned to mouse aortic smooth muscle cells (MOVAS). Necroptosis was induced with 30 ng/mL TNFα plus 60 µM zVAD and cell lysates were prepared 0, 1, 3, or 6 h after the treatment. MLKL phosphorylation on Ser 345 and CaMKII phosphorylation on Thr 287 were evaluated by Western blotting. As shown in Figure 2A–D, levels of both phospho-MLKL and phospho-CaMKII increased after 3 h of necroptotic stimulus. 

As RIPK3-CaMKIIδ interactions were reported to be involved in the induction of necroptosis in cell types such as cardiomyocytes and oligodendrocytes [9,12,22], we further investigated the role of CaMKII in the necroptotic pathway within SMCs. The RIPK1/RIPK3 dual inhibitor GSK’074 completely abolished CaMKII phosphorylation in response to necroptosis (Figure 2E,F). Consistent with prior publications, including those from our own group [5], necroptosis induction caused protein complex formation between RIPK1 and RIPK3. However, immunoprecipitations with either anti-RIPK3 or anti-CaMKII failed to detect complex formations between RIPK3 and CaMKII in necroptotic SMCs (Figure 2G,H).

### 3.2. SMC Necroptosis Requires MLKL

We silenced MLKL with four siRNAs targeting different regions of the Mlkl mRNA. Compared with the scramble control (siNeg), all siRNAs decreased the protein level of MLKL by greater than 80% (Figure 3A,B). In cells that were treated with scramble siRNA, TNFα+zVAD treatment increased the percentage of Annexin V and 7-AAD double positive population from 3.56% to 75.6% (Figure 3C,D). Knocking down MLKL diminished the necroptosis response, as evidenced by a reduction in 7-AAD positive cell populations to less than 25% (Figure 3C,D). 

### 3.3. CaMKII Activation Contributes to SMC Necroptosis

The CaMKII inhibitor myristoylated autocamtide-2-related inhibitory peptide (Myr-AIP) attenuated CaMKII phosphorylation in cells treated with TNFα+zVAD (Figure 4A,B). Inhibition of CaMKII activity with Myr-AIP caused a moderate but statistically significant reduction in necroptosis compared with control (Figure 4C,D). We further examined the role of CaMKII in SMC necroptosis by focusing on CaMKIIδ (encoded by Camk2d), the most abundant isoform expressed in SMCs [23]. We used four siRNAs targeting different regions of the Camk2d mRNA. Compared with the scramble control, all siRNAs showed >80% RNA interference efficiency, and decreased total CaMKII levels to less than 50% (Figure 4E–G). Among these siRNAs, three out of four significantly reduced necroptosis in cells treated with TNFα+zVAD (Annexin V^+^ 7-AAD^+^ population: scramble control 38.83 ± 0.61%, siRNA #1 27.85 ± 1.22%, siRNA #2 22.57 ± 1.01%, siRNA #3 21.78 ± 2.30%, siRNA #4 37.28 ± 5.72%) (Figure 4H,I).

### 3.4. CaMKII Activation Is Downstream of RIPK1/RIPK3-MLKL Pathway in SMC Necroptosis

Having established that both MLKL and CaMKII contribute to SMC necroptosis in a RIPK3 dependent manner, we sought to determine the relationship between MLKL and CaMKII during necroptosis. As shown in Figure 5A,B, CaMKII phosphorylation was significantly diminished in Mlkl siRNA-treated cells. In contrast, three out of four siRNAs against Camk2d failed to affect MLKL expression, phosphorylation, oligomerization, or trafficking (Figure 5C–I, Appendix A). Similarly, silencing Camk2d produced no effects on RIPK1 or RIPK3 expression (Appendix A). 

## 4. Discussion

Despite increasing appreciation of the role that necroptosis plays in human diseases, the signaling steps that drive necroptosis downstream of RIPK3 remain incompletely understood. In many cell types, the phosphorylation of MLKL by RIPK3 and subsequent membrane translation of MLKL are required for execution of necroptosis [7,8]. The findings by Zhang et al. in myocardial necroptosis [9] raise the possibility that RIPK3-downstream signaling steps might be cell-specific. The idea that RIPK3 signals through CaMKII instead of MLKL in cardiomyocytes and perhaps other cell types is interesting and has also been investigated by Chang et al. who demonstrated elevated CaMKII phosphorylation in acute myocardial infarction and tunicamycin-induced cardiomyocyte necroptosis, alongside unaltered levels of MLKL [12]. In contrast, a study conducted by Zhou et al. showed that in vinblastine-induced rat myocardial injury, both MLKL level and phosphorylation of CaMKII were increased. Furthermore, the MLKL inhibitor necrosulfonamide (NSA) partially inhibited cell death [10]. Yang et al. also found that both MLKL and CaMKII were phosphorylated in tissue samples gathered from mice exposed to a combined model of myocardial ischemia-reperfusion injury and chronic pain [24]. 

While knowledge of the precise steps in the necroptotic signaling pathway within vascular SMCs remains elusive, the involvement of necroptosis in the pathogenesis of vascular diseases has clearly emerged. In atherosclerosis, elevated levels of RIPK1, RIPK3, and phosphorylated MLKL have been detected in unstable atherosclerotic plaques from humans [25,26] and *Ripk3*-deficient mice develop significantly smaller advanced aortic atherosclerotic lesions [27]. We have previously demonstrated that RIPK3-mediated SMC necroptosis contributes to abdominal aortic aneurysm pathophysiology [2]. The current study investigated the signaling steps downstream of RIPK3 in the context of aortic aneurysm. Specifically, our data suggests that both MLKL and CaMKII are activated in a CaCl_2_-induced mouse AAA model. The absence of activation of MLKL or CaMKII in *Ripk3*-deficient mice shows that both signaling molecules are downstream from RIPK3. 

We were surprised by the robust effect of *Mlkl* knockdown on SMC necroptosis because we had expected similar findings in SMCs to those reported in cardiomyocytes. The fact that four distinct siRNAs comparably inhibited necroptosis strongly suggests that MLKL is necessary for necroptosis of vascular SMCs. To the best of our knowledge, this is the first study that clearly establishes MLKL as required in necroptosis signaling within vascular SMCs. In the context of atherosclerosis, Rasheed et al. showed that knockdown of *Mlkl* using antisense oligonucleotides in ApoE null mice reduced lesion necrotic core size [28]. While the origin and role of MLKL in atherosclerosis remain elusive, Lin et al. demonstrated that RIPK3 contributes to atherogenesis primarily through bone marrow-derived cells [27].

Using an inhibitor against RIPK1 and RIPK3, we confirmed in cultured vascular SMCs that RIPK1/RIPK3 are required for CaMKII activation. However, our co-immunoprecipitation studies did not detect an interaction between RIPK3 and CaMKII during SMC necroptosis, suggesting that CaMKII may not act immediately downstream from RIPK3. This notion is validated by the MLKL knockdown studies that showed the requirement of MLKL for CaMKII activation. In contrast to our findings, Zhang et al. showed that RIPK3 directly binds to and phosphorylates CaMKII at Thr287 [9]. Similarly, Qu et al. reported increased interaction between RIPK3 and CaMKII in oligodendrocyte progenitor cell necroptosis induced by oxygen-glucose deprivation plus caspase inhibitor zVAD [22]. The origin of the discrepancy between our findings and those reported in the literature regarding RIPK3-CaMKII interactions is unclear. While the lack of apparent RIPK3-CaMKII interactions in our co-immunoprecipitation assay could theoretically be attributed to a technical issue, this seems unlikely given the immunoprecipitation was performed successfully with two different antibodies and included a positive control. It is therefore plausible to postulate that the necroptotic signaling steps might be different depending on cell types and necroptosis stimuli.

CaMKII is a serine/threonine protein kinase whose function has been extensively studied in the brain and myocardium; however, its role in vascular SMCs remains unclear. Among the four isoforms of CaMKII, CaMKIIδ is the most abundant isoform expressed by SMCs [23]. Our data showed that siRNA silencing *Camk2d* caused >80% reduction in the CaMKIIδ protein level, and >50% reduction in total CaMKII level. Interestingly, such efficient silencing of *Camk2d* only partially inhibited SMC necroptosis. We speculate that other CaMKII isoforms may also be involved in SMC necroptosis. Alternatively, RIPK3-MLKL may cause cell death through pathways parallel to CaMKII.

Our results showed that silencing *Mlkl* inhibited CaMKII phosphorylation in SMC necroptosis, indicating CaMKII is downstream from MLKL. Since MLKL is a pseudokinase without catalytic function, we speculate that MLKL indirectly phosphorylates CaMKII. This notion is supported by the lack of interaction between MLKL and CaMKII (data not shown) in SMCs. In TNF-induced necroptosis, oligomerized MLKL translocates to cell plasma membrane where it triggers calcium influx through transient receptor potential cation channel subfamily M member 7 (TRPM7) [29]. It is possible that elevated intracellular calcium contributes to autophosphorylation of CaMKII on Thr287 [23].

There are several limitations in this study. First, we examined phosphorylation of CaMKII on Thr287 as an index of CaMKII activation. The oxidation of CaMKII, another activating event, was not evaluated. We observed one out of the four siRNAs against CaMKIIδ increased MLKL phosphorylation and oligomerization without altering MLKL level. Using NCBI BLAST, we predicted nine unique potential off-targets of this siRNA (*Dysf*, *Gm35315*, *Gm41607*, *Gm7697*, *Gtf2a2*, *LOC118567551*, *Rnf146*, *Washc2*, and *Zfp704*); however, none of these putative off-targets was reported to affect MLKL level or function. We suspect that the inability of this siRNA to inhibit necroptosis may due to uninvestigated effects of these genes, or unidentified off-targets of this siRNA. Moreover, our data showed that both MLKL and CaMKII were phosphorylated in a CaCl_2_-induced murine AAA model, indicating MLKL and CaMKII were likely involved in AAA pathogenesis. Further evaluation using *Mlkl*- or *Camk2d*-deficient mice in AAA models will be highly informative.

In conclusion, our data demonstrates that both MLKL and CaMKII are involved in SMC necroptosis in vitro and in vivo. Silencing *Mlkl* inhibited phosphorylation of CaMKII, indicating TNFα plus zVAD-induced SMC necroptosis is likely mediated by RIPK3-MLKL-CaMKII signaling axis.

## Figures and Tables

**Figure 1 cells-10-02397-f001:**
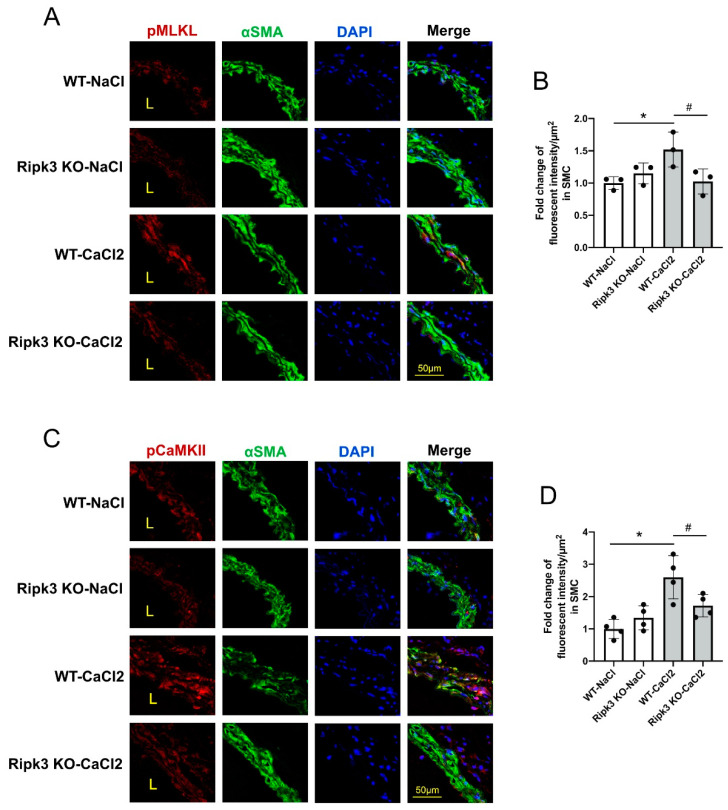
Elevated phosphorylation of MLKL and CaMKII in a murine CaCl_2_-induced AAA model. *Ripk3* wildtype (WT) and knockout (KO) mice were subjected to AAA induction by perivascular treatment with CaCl_2_ or NaCl (sham group). Four days after AAA induction, treated abdominal aortic segments were harvested for cross sections. (**A**,**C**) Anti-alpha smooth muscle actin (green) was used to identify medial smooth muscle cells and DAPI was used to stain nuclei. Representative images of immunostaining of phospho-MLKL Ser345 (red) are shown in panel A and phospho-CaMKII Thr287 (red) in panel C. (**B**,**D**) Quantification of fluorescent intensity of phospho-MLKL Ser345 (**B**) and phospho-CaMKII Thr287 (**D**) is presented relative to the medial layer area. *n* = 3 and 4 for each group in (**B**) and (**D**), respectively. Data were presented as mean ± SD. One-way ANOVA was performed in (**B**) and (**D**). * *p* < 0.05, compared with WT NaCl group; ^#^ *p* < 0.05, compared with WT CaCl_2_ group.

**Figure 2 cells-10-02397-f002:**
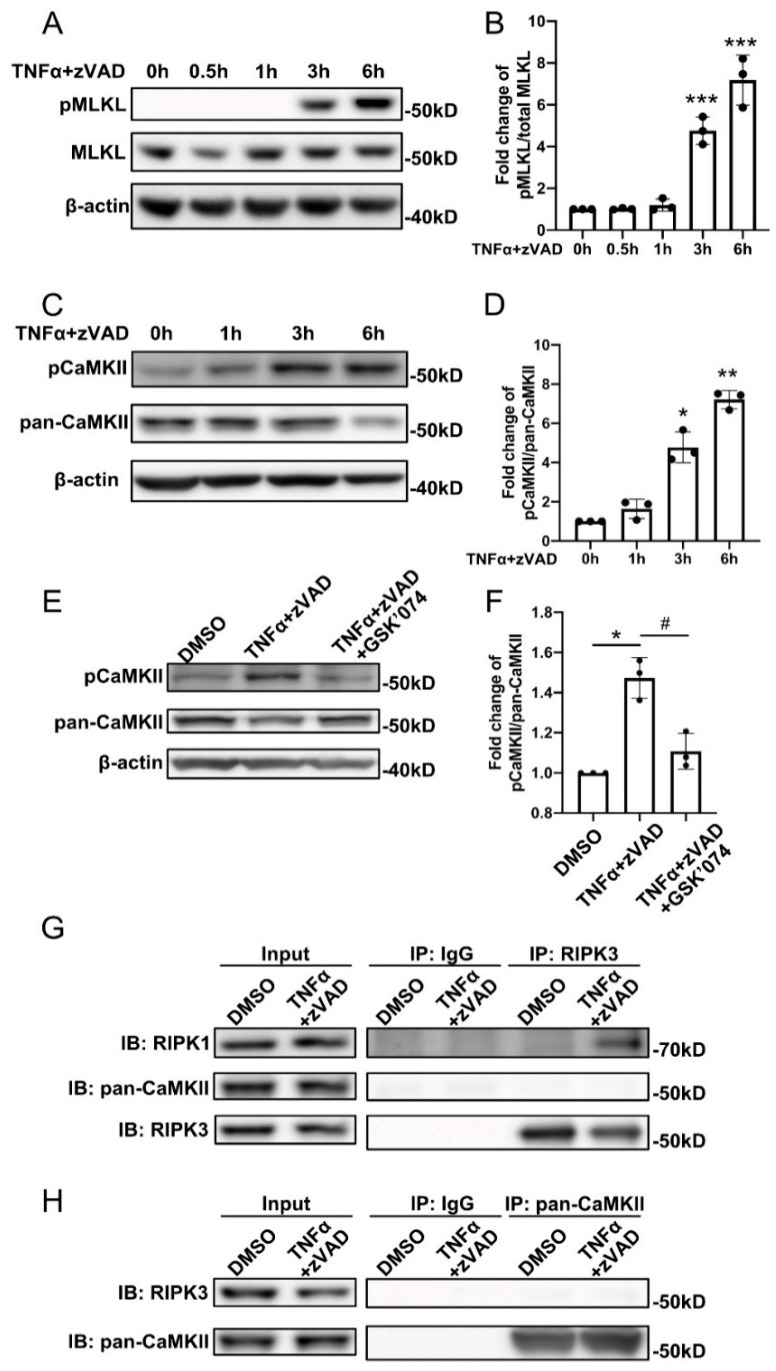
Both MLKL and CaMKII are activated during SMC necroptosis. (**A**,**C**) Mouse aortic smooth muscle cells (MOVAS) were treated with 30 ng/mL TNFα plus 60 µM zVAD for the indicated time period. Cells were lysed in RIPA buffer and analyzed by Western blotting with the indicated antibodies. (**B**) Quantification of (**A**). *n* = 3 for each timepoint. (**D**) Quantification of (**C**). *n* = 3 for each timepoint. (**E**) MOVAS were treated with 30 ng/mL TNFα plus 60 µM zVAD for 6 h in the presence or absence of 10 nM GSK’074. Cells were lysed in RIPA buffer and analyzed by Western blotting with the indicated antibodies. (**F**) Quantification of (**E**). *n* = 3. (**G**,**H**) MOVAS cells were treated with 30 ng/mL TNFα plus 60 μM zVAD for 6 h. Cell lysates were immunoprecipitated with anti-IgG, anti-RIPK3 (**G**), or anti-pan CaMKII (**H**) antibodies followed by Western blotting analysis with the indicated antibodies. Data were presented as mean ± SD of at least three independent experiments. One-way ANOVA was performed in (**B**,**D**,**F**). * *p* < 0.05, ** *p* < 0.01, *** *p* < 0.001. In (**F**), * *p* < 0.05, compared with DMSO group; ^#^ *p* < 0.05, compared with TNFα+zVAD group.

**Figure 3 cells-10-02397-f003:**
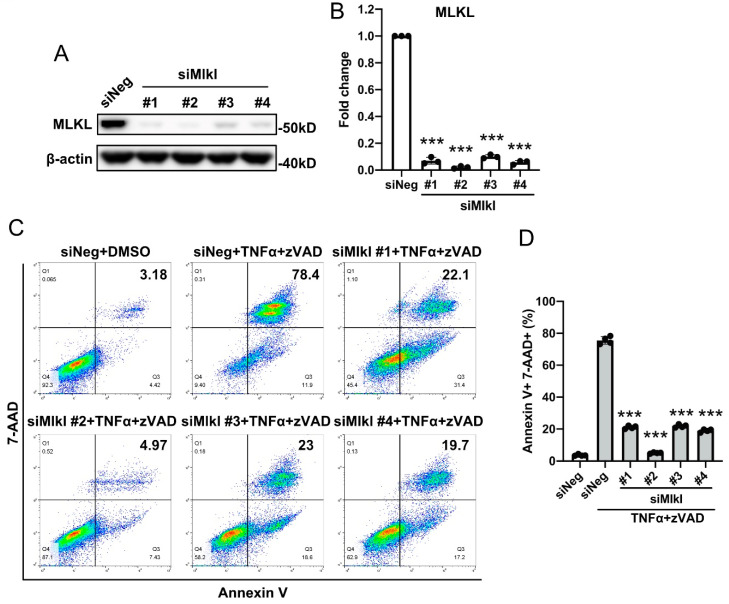
MLKL plays an essential role in SMC necroptosis. MOVAS cells were transfected with siRNAs against *Mlkl* for 48 h. (**A**,**B**) Validation of siRNA knockdown by Western blotting. (**C**) Following *Mlkl* knockdown, MOVAS cells were treated with 30 ng/mL TNFα plus 60 µM zVAD for 6 h. Cells were then stained with 7-AAD and Annexin V, and analyzed by flow cytometry. Necrotic cells were identified as 7-AAD^+^ Annexin V^+^. (**D**) Quantification of (**C**). Data were presented as mean ± SD of at least three independent experiments. One-way ANOVA was performed in (**B**,**D**). *** *p* < 0.001.

**Figure 4 cells-10-02397-f004:**
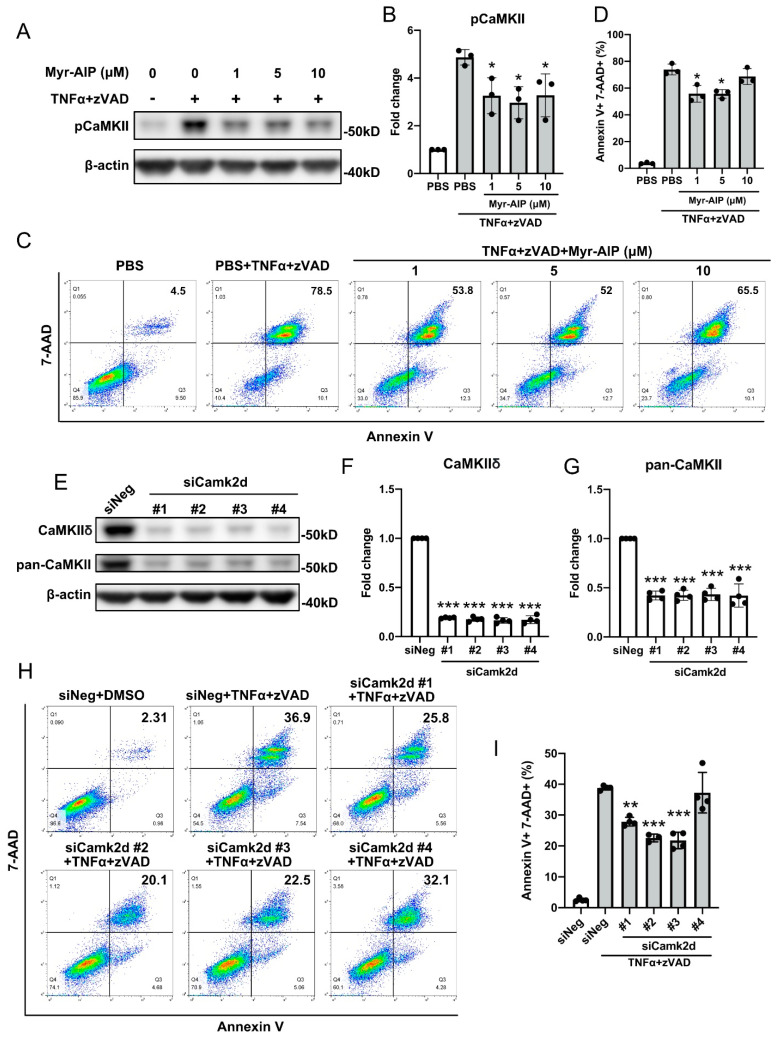
CaMKII is partially required for SMC necroptosis. (**A**–**D**) MOVAS cells were pretreated with indicated doses of CaMKII inhibitor myristoylated autocamtide-2-related inhibitory peptide (Myr-AIP) for 1 h, followed by 6 h of treatment with 30 ng/mL TNFα plus 60 µM zVAD. (**A**) Cells were lysed in RIPA buffer and subjected to Western blotting analysis with the indicated antibodies. (**C**) Cells were stained with 7-AAD and Annexin V, and subjected to flow cytometry analysis. Necrotic cells were identified as 7-AAD^+^ Annexin V^+^. (**B**) was quantification of (**A**). (**D**) was quantification of (**C**). (**E**) MOVAS cells were transfected with siRNAs against *Camk2d* for 48 h. Cells were lysed in RIPA buffer and subjected to Western blotting analysis with the indicated antibodies. (**F**,**G**) Quantification of (**E**). (**H**) MOVAS cells were transfected with siRNAs against *Camk2d* for 48 h, then treated with 30 ng/mL TNFα plus 60 µM zVAD for 6 h. Cells were then stained with 7-AAD and Annexin V, and subjected to flow cytometry analysis. Necrotic cells were identified as 7-AAD^+^ Annexin V^+^. (**I**) Quantification of (**H**). Data were presented as mean ± SD of at least three independent experiments. One-way ANOVA was performed in (**B**,**D**,**F**,**G**,**I**). * *p* < 0.05, ** *p* < 0.01, *** *p* < 0.001 compared with TNFα plus zVAD treated group (**B**,**D**,**I**) or scrambled siRNA-treated group (**F**,**G**).

**Figure 5 cells-10-02397-f005:**
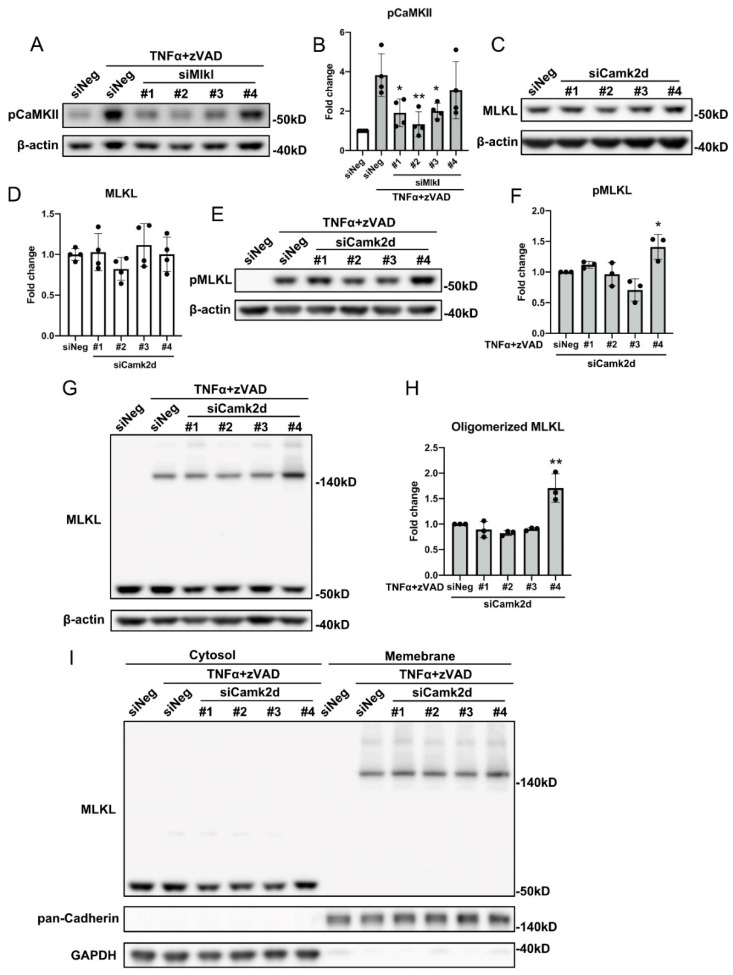
CaMKII activation requires MLKL. (**A**) MOVAS cells were transfected with different siRNAs against *Mlkl* for 48 h, then treated with 30 ng/mL TNFα plus 60 µM zVAD for 6 h. Cells were lysed in RIPA buffer and subjected to Western blotting analysis with the indicated antibodies. (**B**) Quantification of (**A**). (**C**) MOVAS cells were transfected with siRNAs against *Camk2d* for 48 h, cells were lysed in RIPA buffer and subjected to Western blotting analysis with the indicated antibodies. (**D**) Quantification of (**C**). (**E**–**I**) MOVAS cells were transfected with siRNAs against *Camk2d* for 48 h, then treated with 30 ng/mL TNFα plus 60 µM zVAD for 6 h. (**E**,**G**) Cells were lysed in RIPA buffer (**E**) or 1% digitonin (**G**) and were subjected to Western blotting analysis with the indicated antibodies. (**I**) Cytosol and membrane fractions were subjected to Western blotting analysis with the indicated antibodies. (**F**) Quantification of (**E**). (**H**) Quantification of (**G**). Data were presented as mean ± SD of at least three independent experiments. One-way ANOVA was performed in (**B**,**D**,**F**,**H**). * *p* < 0.05, ** *p* < 0.01 compared with TNFα plus zVAD-treated group (**B**,**F**,**H**).

## Data Availability

The data that support the findings of this study are available from the corresponding author upon reasonable request.

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
