# Peer review of "MLKL and CaMKII Are Involved in RIPK3-Mediated Smooth Muscle Cell Necroptosis"

_cells, 2021, doi:10.3390/cells10092397_

Round 1

Reviewer 1 Report

Cell death has been extensively investigated for decades, since serving as an essential process throughout life. It is well known that historically, cell death was classified as distinctive forms: apoptosis, and necrosis. The historical concept of programmed cell death has been associated with apoptosis because it is considered a form of suicide, based on a genetic mechanism. Any cell death other than apoptosis has generally been called “accidental cell death” Necrosis has consequently been described as accidental cell death rather than as the result of definite pathways. It has now been established that necrosis is not an accidental, passive, unregulated form of cell death, but like apoptosis, it can be governed by a “regulated” mechanism, meaning a death with the classic morphological features of necrosis but genetically determined. Since several specialized forms of regulated necrosis have been describe, at a molecular level, the best-characterized pathway of regulated necrosis (RN) is necroptosis. Now, it has been well documented that necroptosis is highly orchestrated regulated necrosis dependent on receptor-interacting protein kinase 3 (RIP3) and mixed lineage kinase domain-like protein (MLKL). It not only involves in the development of organism but also participates in various pathophysiological processes. An accumulating body of evidence has demonstrated that necroptosis contributes to the pathogenesis of numerous diseases and tissue damages.

In this perspective, aim of authors is interestingly, as they examine into the molecular mechanisms that trigger the control of necroptosis in smooth muscle cell. The authors have previously demonstrated that aortic tissue from abdominal aortic aneurysms patients have elevated levels of RIPK3. In mice, inhibition of necroptosis either by gene deletion of Ripk3 or by chemical inhibitors to RIPK1 or both RIPK1 and RIPK3 prevents medial SMC depletion and reduces aortic dilatation. Moreover, since other authors have seen possible involvement of CaMKII in necroptosis, the aim of this study was to examine the role of both MLKL and CaMKII, in smooth muscle cell necroptosis.

The topic of the manuscript is fairly inherent to the aim and scope of the journal “Cells”. With this work, the authors' aim would be to address the elucidation of the signaling pathway that seems to suggest that both MLKL and CaMKII participate in RIPK3-mediated SMC necroptosis.

The design of the study is well done, and the paper clearly written. The experiments are well conducted and organized in the aims of authors, both from an experimental point of view and in terms of the techniques used to confirm the objective to be achieved. The results are well discussed. The conclusions are appropriate.

Some minor correction should be do. All suggestions and requested corrections are provided below.

  1. IN THE “MATERIALS AND METHODS” SECTION:

Subchapter: Flow cytometric analysis

Page 3, lines 117-119: the authors wrote: “After incubation, 400 ml binding buffer was added to each sample. Cells were analyzed using a Becton Dickinson Biosciences FACSCalibur”

  • Authors should specify the minimum number of cells analyzed per sample
  1. IN THE “RESULTS” SECTION:

In the legends of figures 1 and 2: the authors wrote: “*,#p<0.05”

  • Why do the authors use two different symbols to indicate the same significance? If it is the same, only one relative symbol should be used, but if each symbol indicates a different significance, the authors should add and indicate the missing one.

Subchapter: 3.3 CaMKII activation contributes to SMC necroptosis

Page 8, lines 242-243: the authors wrote: “moderate but significant reduction”

  • the two words put in this way are in contrast, it is necessary to write better. For example: “Inhibition of CaMKII activity with Myr-AIP caused a moderate but statistically significant reduction in necroptosis compared to control”
  1. FIGURES

With regard to the figures, some the graphs are not clear: in particular Figure 3D, 4I

  • Authors must improve graphics and quality.
  1. IN THE “DISCUSSION” SECTION:

Page 13, line 407: the authors wrote: “For an unknown reason (maybe an off-target effect),”

  • Have the authors checked off-targets? or are they just speculating? Authors for completeness of analysis must also check the eventual off-targets effect to strengthen and be sure of the result.

Reviewer 2 Report

1. in line 38/39 please provide the bibliography showed the facts of demonstrated RIPK3 elevated levels in AAA patients or indicate that as unpublished data.

2. in lines 78 and 82 please provide of  size of group ( n= ...) CaCl and sham,  respectively. 
